# Effects of abscisic acid on growth and selenium uptake in medicinal plant *Perilla frutescens*

**Renyan Liao** [ID]*

College of Traditional Chinese Medicine and Rehabilitation, Ya'an Polytechnic College, Ya'an, Sichuan, China

* 51699474@qq.com

**Data Availability Statement:** All relevant data are within the paper and its Supporting information files.

**Funding:** The author(s) received no specific funding for this work.

## Abstract

The aim of the present study was to explore the effects of abscisic acid (ABA) on growth and selenium (Se) absorption of the medicinal plant, *Perilla frutescens*. A pot experiment was conducted to evaluate the effects of different ABA concentrations (0, 1, 5, 10 and 20 μmol/L) on the physiological characteristics and Se absorption capacity of *P. frutescens*. Application of 5, 10 and 20 μmol/L ABA increased the shoot biomass of *P. frutescens*, and only 5 and 10 μmol/L ABA increased the root biomass. Application of 5, 10, and 20 μmol/L ABA increased the contents of photosynthetic pigments (chlorophyll *a*, chlorophyll *b*, total chlorophyll, and carotenoid), superoxide dismutase activity, peroxidase activity, and soluble protein content of *P. frutescens*, and decreased the malondialdehyde content in *P. frutescens*. Only 5 and 10 μmol/L ABA used in the present study increased the catalase activity of *P. frutescens*. For the Se uptake, only 5 μmol/L ABA increased the Se content, Se extraction and Se bioconcentration factor of both roots and shoots. The findings of the present study indicate that 5 and 10 μmol/L ABA promotes the growth of *P. frutescens*, whereas 5 μmol/L ABA enhances the Se accumulation capacity in *P. frutescens*.

## Introduction

Selenium (Se) is an essential trace element for human growth and development. It plays an important role in preventing diseases, improving health and slowing down aging [1, 2]. Findings from a previous survey showed that the daily Se intake *per capita* in China was lower relative to the standard recommended by the World Health Organization [3]. One of the main sources of Se in human body is plant based diet [4]. Absorption of Se by plants is a safe, economic and effective way to convert Se in the soil into organic Se [5]. Therefore, several studies are exploring Se enrichment in plants.

Abscisic acid (ABA) was initially considered an inhibitory plant hormone [6]. However, recent studies show that ABA widely regulates growth and development processes of plants, such as seed germination, flowering and stomatal opening and closing [7, 8]. Moreover, it plays an important role in abiotic stress response in plants [9]. Studies report that appropriate

**Competing interests:** The authors have declared that no competing interests exist.

**Abbreviations:** ABA, Abscisic acid; BCF, Bioconcentration factor; CAT, Catalase; MDA, Malondialdehyde; POD, Peroxidase; Se, Selenium; SOD, Superoxide dismutase; TAF, Translocation accumulation factor; TF, Translocation facto.

concentrations of ABA can improve adaptation of plants to drought conditions [10], low temperature [11], and saline alkali conditions [12]. In addition, ABA affects absorption of elements or ions by plants. Application of appropriate concentrations of ABA regulates absorption of sodium, potassium, calcium and magnesium ions by *Toona sinensis* and alfalfa, thus modulating their ion balance [13, 14]. Furthermore, ABA within a certain concentration range can reduce cadmium content in the shoots of rice [15] and *Brassica campestris* [16]. Notably, ABA improves cadmium absorption by some accumulator and hyperaccumulator plants such as *Stellaria media* and *Solanum photeinocarpum* [8, 17]. These results indicate that ABA modulates absorption of elements from the soil by plants.

*Perilla frutescens* is an annual herb of Labiatae family. It is widely distributed in China. It is a traditional medicinal and edible plant [18, 19], rich in fiber, carotene and mineral elements, and has great nutritional value [20]. A previous study reported that *P. frutescens* has strong Se accumulation ability from soil [21]. However, the accumulation rate is lower compared with that of other Se hyperaccumulators such as *Cardamine hupingshanensis* [22] and *Cardamine enshiensis* [23]. Therefore, the present study sought to explore the effects of ABA on growth and Se absorption characteristics of *P. frutescens* to improve its Se accumulation ability. The aim of this study was to explore the most effective ABA concentrations in promoting growth and Se uptake of *P. frutescens*, to provide a basis for *P. frutescens* production development.

## Materials and methods

### Materials

*P. frutescens* seeds were collected from fields around the Ya'an Polytechnic College, located in Yucheng District, Ya'an City, Sichuan Province, China, in October 2020. Seeds were air-dried and stored at 4°C. Inceptisol soil samples (purple soil according to the Genetic Soil Classification of China) were collected from the farmland fields around Ya'an Polytechnic College in February 2021. Basic properties of the soil samples were as follows: pH value, 7.71; organic matter content, 13.45 g/kg; alkaline nitrogen content, 102.35 mg/kg; available phosphorus content, 68.45 mg/kg; available potassium content, 48.49 mg/kg; and total Se content, 0.02 mg/kg.

### Experimental design

Pot experiments were conducted at Ya'an Polytechnic College from February 2021 to May 2021. Air-dried, crushed and sieved (5 mm) soil samples (3.0 kg) were placed in a plastic pot of 15 cm × 18 cm (height × diameter) in February 2021. $Na_2SeO_3$ solution was added to the soil samples and mixed thoroughly to achieve a final Se concentration at 5 mg/kg [21]. Soil moisture was maintained at 80% of the field capacity for four weeks. Whole and undamaged *P. frutescens* seeds were selected and placed in an incubator for germination. Four uniform seedlings were transplanted into each pot, after the plants attained three pairs of expanded true leaves. Seedlings were then watered every day to maintain the soil moisture content at 80% of the field capacity until the plants were harvested. Gradient concentrations of ABA solution [0 (the control), 1, 5, 10 and 20 μmol/L] were sprayed on the leaves and stems of plants a week after transplantation. Spraying criteria were based on a layer of small water droplets uniformly attached to the leaves during each phase of spraying (about 15 mL solution for each pot). Each treatment was conducted in triplicates. ABA solution [0 (the control), 1, 5, 10 and 20 μmol/L] further sprayed a week later after the first spraying phase.

Mature leaves of each plant (the fourth pair of leaves from the top) were collected a month after the first application of ABA. These leaf samples were used to determine content of photosynthetic pigments (chlorophyll *a*, chlorophyll *b*, total chlorophyll and carotenoid), antioxidant enzyme [superoxide dismutase (SOD), peroxidase (POD) and catalase (CAT)] activity,

soluble protein content, and malondialdehyde (MDA) content. Leaf samples were weighed (0.200 g) and soaked in ethanol and acetone at a volume ratio of 1:1 for extraction. A spectrophotometer was then used to measure the absorbance and the content of photosynthetic pigments was determined [24]. Extraction of enzyme solution was performed at 4˚C. Fresh leave tissues (1.0 g) were homogenized in 6 mL of potassium phosphate buffer (pH 7.0) and centrifuged. The supernatant was used for determination of enzyme activity, soluble protein content, and MDA content. SOD activity was determined through nitroblue tetrazolium photoreduction method, POD activity was determined following guaiacol colorimetric method, CAT activity was determined through UV spectrophotometry method, Soluble protein content was determined by Coomassie brilliant blue staining method, and MDA content was determined according to thiobarbital acid described previously [24]. Then the whole plant was harvested, washed with deionized water and dried at 75˚C to a constant weight. Biomass (dry weight) of roots, stems, and leaves was determined by electronic balance (precision 0.01g). Dried plant tissues were ground, and were for determination of Se content. Plant samples (0.5 g) were digested with $HNO_3/HClO_4$ (4:1, v/v) and Se content determined using ICP spectrometer (iCAP 6300, Thermo Scientific, Waltham, MA, USA). Other parameters related to biomass and Se content were determined as follows: shoot biomass = stem biomass + leaf biomass; root/shoot ratio = root biomass/shoot biomass; Se content in shoots = (Se content in stems × stem biomass + Se content in leaves × leaf biomass)/ shoot biomass; Se translocation factor (TF) = Se content in shoots/ Se content in roots [25]; Se extraction = Se content in plant × plant biomass [26]; Se translocation amount factor (TAF) = Se extraction by shoots/Se extraction by roots [27]; Se bioconcentration factor (BCF) = Se content in plant/ Se concentration in soil [25].

## Statistical analyses

Statistical analyses were conducted using SPSS 17.0 software. Differences between groups were determined by one-way ANOVA. The Duncan's Multiple Range Test was conducted at 95% confidence level. A regression analysis was used to analyze the relationship between ABA concentration and biomass, Se content or Se extraction. The relationships among all items were analyzed by Pearson correlation.

## Results

### Biomass of *P. frutescens*

The findings showed that use of 1 and 20 μmol/L ABA had no significant ($P > 0.05$) effect on the root biomass of *P. frutescens* compared with the control. Administration of 5 and 10 μmol/ L ABA significantly increased the root biomass compared with the control (Table 1). Administration of 5, 10 and 20 μmol/L ABA increased the stem, leaf and shoot biomass of *P. frutescens*, whereas 1 μmol/L ABA had no significant ($P > 0.05$) effect on the stem, leaf and shoot biomass compared with the control. Administration of ABA at 5 μmol/L, increased the root, stem, leaf and shoot biomass of *P. frutescens* by 14.52%, 24.88%, 11.50%, and 17.20%, respectively, compared with those of the control. Regression analysis showed that the root biomass had a quadratic polynomial regression relationship with ABA concentration (x: ABA concentration; y: root biomass; y = -0.002x² + 0.050x + 2.447, R² = 0.444, $P = 0.030$), and shoot biomass also had a quadratic polynomial regression relationship with ABA concentration (x: ABA concentration; y: shoot biomass; y = -0.006x² + 0.134x + 5.091, R² = 0.684, $P = 0.000$). Comparison of the effect of ABA treatment on the root/shoot ratio of *P. frutescens* showed no significant difference ($P > 0.05$) among the different ABA treatments.

**Table 1. Biomass of *P. frutescens*.**

| ABA | Roots | Stems | Leaves | Shoots | Root/shoot ratio |
|---|---|---|---|---|---|
| (μmol/L) | (g/plant) | (g/plant) | (g/plant) | (g/plant) | |
| 0 | 2.41±0.13c | 2.13±0.02d | 2.87±0.11c | 5.00±0.12c | 0.483±0.031a |
| 1 | 2.48±0.03bc | 2.24±0.11cd | 2.97±0.02bc | 5.21±0.09bc | 0.477±0.011a |
| 5 | 2.76±0.16a | 2.66±0.10a | 3.20±0.11a | 5.86±0.21a | 0.470±0.029a |
| 10 | 2.63±0.13ab | 2.47±0.01b | 3.17±0.04a | 5.64±0.03a | 0.466±0.023a |
| 20 | 2.53±0.02bc | 2.31±0.01c | 3.09±0.12ab | 5.41±0.13b | 0.469±0.009a |

Different lowercase letters within a column indicate significant differences based on one-way analysis of variance (ANOVA) with the Duncan's Multiple Range Test ($P < 0.05$). Shoot biomass = stem biomass + leaf biomass, root/shoot ratio = root biomass/shoot biomass.

## Photosynthetic pigment content of *P. frutescens*

Treatment with 1, 5, 10, and 20 μmol/L ABA significantly increased the contents of chlorophyll *a*, chlorophyll *b*, total chlorophyll and carotenoid in *P. frutescens* compared with the control (Table 2). Administration of 5 μmol/L ABA increased the chlorophyll *a*, chlorophyll *b*, total chlorophyll and carotenoid contents by 21.45%, 28.00%, 22.97%, and 16.48%, respectively relative to those of the control. Administration of 1, 5, 10, and 20 μmol/L ABA exhibited no significant ($P > 0.05$) difference in chlorophyll *a/b* compared with the control.

## Antioxidant enzyme activity, soluble protein content, and MDA content of *P. frutescens*

Administration of the various concentrations of ABA increased the SOD activity of *P. frutescens* compared with the control (Table 3). Treatment with 1 μmol/L ABA had no significant ($P > 0.05$) effects on the POD activity and soluble protein content of *P. frutescens* compared with the control, while use of 5, 10, and 20 μmol/L ABA increased. Treatment with 1 and 20 μmol/L ABA had no significant ($P > 0.05$) effects on the CAT activity of *P. frutescens* compared with the control, while use of 5 and 10 μmol/L ABA increased. Administration of the various concentrations of ABA decreased the MDA content in *P. frutescens* compared with the control.

## Se content in *P. frutescens*

Administration of 5 μmol/L ABA increased Se content in roots and shoots of *P. frutescens* by 16.10% and 11.30%, respectively, compared with the control. On the contrary, 10 and

**Table 2. Photosynthetic pigment content in *P. frutescens*.**

| ABA | Chlorophyll *a* | Chlorophyll *b* | Total chlorophyll | Chlorophyll | Carotenoid |
|---|---|---|---|---|---|
| (μmol/L) | (mg/g) | (mg/g) | (mg/g) | *a/b* | (mg/g) |
| 0 | 3.31±0.09c | 1.00±0.02d | 4.31±0.11d | 3.31±0.10ab | 0.704±0.010c |
| 1 | 3.57±0.08b | 1.06±0.03c | 4.63±0.10c | 3.37±0.11a | 0.725±0.022c |
| 5 | 4.02±0.13a | 1.28±0.03a | 5.30±0.16a | 3.14±0.12b | 0.820±0.015a |
| 10 | 3.92±0.10a | 1.25±0.04a | 5.17±0.13a | 3.13±0.10b | 0.787±0.026ab |
| 20 | 3.72±0.12b | 1.20±0.02b | 4.92±0.13b | 3.11±0.11b | 0.771±0.027b |

Different lowercase letters within a column indicate significant differences based on one-way analysis of variance (ANOVA) with the Duncan's Multiple Range Test ($P < 0.05$).

**Table 3. Antioxidant enzyme activity, soluble protein content, and MDA content of *P. frutescens*.**

| ABA | SOD activity | POD activity | CAT activity | Soluble protein | MDA content |
|---|---|---|---|---|---|
| (µmol/L) | (U/g) | (U/g/min) | (U/g/min) | content (mg/g) | (µmol/kg) |
| 0 | 152.17±3.74d | 119.44±3.70d | 6.09±0.23c | 37.18±1.22c | 7.11±0.14a |
| 1 | 182.92±2.80c | 122.35±9.85d | 6.17±0.22bc | 37.96±0.88c | 6.72±0.24b |
| 5 | 242.38±5.29a | 276.07±11.93a | 6.84±0.18a | 47.39±1.83a | 5.85±0.12c |
| 10 | 213.84±9.96b | 188.19±7.61b | 6.44±0.11b | 45.77±1.76ab | 5.65±0.19c |
| 20 | 209.64±7.48b | 168.54±7.04c | 6.31±0.01bc | 43.77±1.51b | 6.43±0.16b |

Different lowercase letters within a column indicate significant differences based on one-way analysis of variance (ANOVA) with the Duncan's Multiple Range Test ($P < 0.05$). SOD: Superoxide dismutase; POD: Peroxidase; CAT: Catalase; MDA: Malondialdehyde.

20 µmol/L ABA decreased the Se contents in roots and shoots (Table 4). Notably, administration of 1 µmol/L ABA increased the Se contents in roots and had no significant ($P > 0.05$) effect on the Se contents in shoots compared with the control. Treatment with 1 and 5 µmol/L ABA increased the Se contents in stems of *P. frutescens* relative to the control, whereas use of 10 and 20 µmol/L ABA had no significant effect ($P > 0.05$) or decreased. Analysis of Se content in leaves showed that 1 and 5 µmol/L ABA had no significant effects ($P > 0.05$), whereas treatment with 10 and 20 µmol/L ABA decreased the Se content in leaves. Regression analysis showed that the root Se content had a linear regression relationship with ABA concentration (x: ABA concentration; y: root Se content; y = -0.810x + 65.175, $R^2$ = 0.559, $P$ = 0.001), and the shoot Se content also had a linear regression relationship with ABA concentration (x: ABA concentration; y: shoot Se content; y = -0.362x + 24.413, $R^2$ = 0.721, $P$ = 0.000). The results showed no significant ($P > 0.05$) differences in the TF of *P. frutescens* among the different concentrations of ABA.

## BCF of *P. frutescens*

Administration of 1 and 5 µmol/L ABA increased the root and stem BCFs of *P. frutescens* relative to the control, whereas other ABA concentrations showed no significant ($P > 0.05$) effect or decreased (Table 5). The results showed that 1 and 5 µmol/L ABA had no significant ($P > 0.05$) effects on the leaf BCF compared with the control, whereas use of 10 and 20 µmol/L ABA decreased. Treatment with 5 µmol/L ABA increased the shoot BCF compared with the control, whereas other ABA concentrations had no significant ($P > 0.05$) effect or decreased.

**Table 4. Se content in *P. frutescens*.**

| ABA | Roots | Stems | Leaves | Shoots | TF |
|---|---|---|---|---|---|
| (µmol/L) | (mg/kg) | (mg/kg) | (mg/kg) | (mg/kg) | |
| 0 | 60.44±1.31c | 16.07±0.49c | 27.71±1.19a | 22.75±0.96b | 0.376±0.018a |
| 1 | 64.52±1.28b | 17.94±0.86b | 28.94±1.54a | 24.20±1.10ab | 0.375±0.020a |
| 5 | 70.17±3.79a | 20.17±0.45a | 29.61±0.82a | 25.32±0.64a | 0.362±0.021a |
| 10 | 52.68±0.93d | 16.29±0.50c | 22.49±1.28b | 19.77±0.91c | 0.376±0.021a |
| 20 | 48.91±1.65e | 14.14±0.63d | 19.12±0.73c | 17.00±0.20d | 0.348±0.011a |

Different lowercase letters within a column indicate significant differences based on one-way analysis of variance (ANOVA) with the Duncan's Multiple Range Test ($P < 0.05$). Se content in shoots = (Se content in stems × stem biomass + Se content in leaves × leaf biomass)/ shoot biomass, Se translocation factor (TF) = Se content in shoots/ Se content in roots.

**Table 5. BCF of *P. frutescens*.**

| ABA (μmol/L) | Root BCF | Stem BCF | Leaf BCF | Shoot BCF |
|---|---|---|---|---|
| 0 | 12.09±0.26c | 3.21±0.10c | 5.54±0.24a | 4.55±0.19b |
| 1 | 12.90±0.26b | 3.59±0.17b | 5.79±0.31a | 4.84±0.22ab |
| 5 | 14.03±0.76a | 4.03±0.09a | 5.92±0.16a | 5.07±0.13a |
| 10 | 10.54±0.19d | 3.26±0.10c | 4.50±0.26b | 3.95±0.18c |
| 20 | 9.78±0.33d | 2.83±0.13d | 3.82±0.15c | 3.40±0.04d |

Different lowercase letters within a column indicate significant differences based on one-way analysis of variance (ANOVA) with the Duncan's Multiple Range Test ($P < 0.05$). Se bioconcentration factor (BCF) = Se content in plant/ Se concentration in soil.

## Se extraction by *P. frutescens*

The findings showed that 1 and 5 μmol/L ABA increased the Se extraction by roots of *P. frutescens* by 9.95%, and 32.43%, respectively, relative to the control, whereas other ABA concentrations had no effect ($P > 0.05$) significant or decreased (Table 6). Administration of 1, 5, and 10 μmol/L ABA increased the Se extraction by stems relative to the control, whereas use of 20μmol/L ABA showed no significant ($P > 0.05$) effect. Notably, only 5 μmol/L ABA increased the Se extractions by leaves and shoots by 19.30% and 30.59%, respectively, whereas other ABA concentrations had no significant ($P > 0.05$) effect or decreased. Regression analysis showed that the root Se extraction had a linear regression relationship with ABA concentration (x: ABA concentration; y: root Se extraction; $y = -1.853x + 165.674$, $R^2 = 0.320$, $P = 0.028$), and the shoot Se extraction also had a linear regression relationship with ABA concentration (x: ABA concentration; y: shoot Se extraction; $y = -1.648x + 130.249$, $R^2 = 0.381$, $P = 0.014$). The results showed no significant ($P > 0.05$) differences in the TAF of *P. frutescens* among the different ABA concentrations.

## Correlation analyses

Both the root biomass and shoot biomass showed a highly significant ($P < 0.01$) positive correlation with the chlorophyll *a* content, chlorophyll *b* content, total chlorophyll content, carotenoid content, POD activity, SOD activity, CAT activity, and soluble protein content, while showed a highly significant ($P < 0.01$) negative correlation with the MDA content (Table 7). Both the root Se content and shoot Se content had no significant ($P \geq 0.05$) correlation with the indicators of biomass, photosynthetic pigment content, antioxidant enzyme activity,

**Table 6. Se extraction by *P. frutescens*.**

| ABA (μmol/L) | Roots (μg/plant) | Stems (μg/plant) | Leaves (μg/plant) | Shoots (μg/plant) | TAF |
|---|---|---|---|---|---|
| 0 | 145.79±4.71c | 34.28±1.37c | 79.50±6.30bc | 113.78±7.61bc | 0.781±0.062a |
| 1 | 160.30±5.28b | 40.32±3.99b | 85.82±4.05b | 126.14±8.04b | 0.788±0.061a |
| 5 | 193.07±2.52a | 53.74±3.27a | 94.84±5.79a | 148.58±9.06a | 0.769±0.037a |
| 10 | 138.60±8.67c | 40.25±1.31b | 71.26±3.23c | 111.52±4.54c | 0.807±0.057a |
| 20 | 123.89±4.20d | 32.70±1.36c | 59.20±4.61d | 91.90±3.24d | 0.742±0.020a |

Different lowercase letters within a column indicate significant differences based on one-way analysis of variance (ANOVA) with the Duncan's Multiple Range Test ($P < 0.05$). Se extraction = Se content in plant × plant biomass, Se translocation amount factor (TAF) = Se extraction by shoots/Se extraction by roots.

**Table 7. Correlations among biomass, Se content, Se extraction, photosynthetic pigment content, antioxidant enzyme activity, soluble protein content, and MDA content.**

| Items | Root biomass | Shoot biomass | Chlorophyll a content | Chlorophyll b content | Total chlorophyll content | Carotenoid content | POD activity | SOD activity | CAT activity | Soluble protein content | MDA content | Root Se content | Shoot Se content | Root Se extraction | Shoot Se extraction |
|---|---|---|---|---|---|---|---|---|---|---|---|---|---|---|---|
| Root biomass | 1 | | | | | | | | | | | | | | |
| Shoot biomass | 0.759** | 1 | | | | | | | | | | | | | |
| Chlorophyll *a* content | 0.785** | 0.884** | 1 | | | | | | | | | | | | |
| Chlorophyll *b* content | 0.754** | 0.901** | 0.917** | 1 | | | | | | | | | | | |
| Total chlorophyll content | 0.744** | 0.919** | 0.936** | 0.979** | 1 | | | | | | | | | | |
| Carotenoid content | 0.726** | 0.834** | 0.870** | 0.944** | 0.930** | 1 | | | | | | | | | |
| POD activity | 0.718** | 0.890** | 0.831** | 0.845** | 0.844** | 0.863** | 1 | | | | | | | | |
| SOD activity | 0.762** | 0.915** | 0.921** | 0.938** | 0.929** | 0.888** | 0.895** | 1 | | | | | | | |
| CAT activity | 0.732** | 0.823** | 0.787** | 0.772** | 0.806** | 0.744** | 0.876** | 0.821** | 1 | | | | | | |
| Soluble protein content | 0.744** | 0.901** | 0.899** | 0.940** | 0.902** | 0.840** | 0.860** | 0.939** | 0.798** | 1 | | | | | |
| MDA content | -0.676** | -0.857** | -0.877** | -0.850** | -0.832** | -0.747** | -0.761** | -0.839** | -0.682** | -0.868** | 1 | | | | |
| Root Se content | 0.170 | 0.162 | 0.023 | -0.088 | 0.020 | 0.077 | 0.337 | 0.096 | 0.304 | -0.093 | 0.061 | 1 | | | |
| Shoot Se content | 0.170 | 0.105 | -0.019 | -0.187 | -0.072 | -0.059 | 0.205 | 0.001 | 0.255 | -0.127 | 0.095 | 0.933** | 1 | | |
| Root Se extraction | 0.537* | 0.440 | 0.333 | 0.224 | 0.313 | 0.354 | 0.580* | 0.387 | 0.558* | 0.219 | -0.212 | 0.922** | 0.862** | 1 | |
| Shoot Se extraction | 0.448 | 0.482 | 0.326 | 0.189 | 0.296 | 0.274 | 0.535* | 0.363 | 0.555* | 0.245 | -0.242 | 0.888** | 0.921** | 0.935** | 1 |

**: Correlation is significant at the 0.01 level (2-tailed).

*: Correlation is significant at the 0.05 level (2-tailed). N = 15. SOD: Superoxide dismutase; POD: Peroxidase; CAT: Catalase; MDA: Malondialdehyde.

soluble protein content, and MDA content. Both the root Se extraction and shoot Se extraction showed a highly significant ($P < 0.01$) positive correlation with the root Se content and shoot Se content. The root Se extraction showed a significant ($0.05 > P \geq 0.01$) positive correlation with the root biomass, POD activity, and CAT activity, while the shoot Se extraction showed a significant ($0.05 > P \geq 0.01$) positive correlation with the POD activity and CAT activity.

## Discussion

Biomass is an important index for determining plant growth. The most direct effect of adversity on plants is to inhibit growth and reduce biomass [28]. ABA promotes the growth of pepper plants under salt stress by increasing the plant height and biomass [29]. Moreover, ABA increases *Foeniculum vulgare* plant height under cadmium stress [30]. Notably, ABA increases the biomass of the potential cadmium hyperaccumulator *Solanum photeinocarpum* under cadmium contaminated soil conditions [8], whereas it decreases the biomass of the cadmium hyperaccumulator *Bidens pilosa* [31]. These findings indicate that ABA has different effects on the growth of various plant species. In the present study, administration of 5 and 10 µmol/L ABA increased the biomass of *P. frutescens*, whereas 1 µmol/L ABA had no significant effect, which is consistent with findings from previous studies [8, 32]. In addition, regression analysis showed that the ABA concentration had a quadratic polynomial regression relationship with the root biomass and shoot biomass of *P. frutescens*. These results indicate that ABA promotes the growth of *P. frutescens* by improving the resistance to soil Se stress [8, 21].

ABA is a hormone involved in regulating senescence of plants. Exogenous ABA accelerates leaf yellowing and chlorophyll degradation by inducing up-regulation of senescence-related genes, thus inducing the senescence of *Capsicum annuum* leaves under dark culture *in vitro* [33]. ABA is an important stress hormone in plants that inhibits the production of chlorophyll in perennial *Lolium perenne* under normal growth, but upregulates the chlorophyll content under drought stress [34]. Moreover, ABA application increases the chlorophyll content in leaves of sweet potato under water stress [35]. Exogenous ABA increases the photosynthetic pigment content in *Lycium barbarum* under NaCl stress, mainly the carotenoid content [36]. Application of 5, 10, and 20 µmol/L ABA in the current study increased chlorophyll *a*, chlorophyll *b*, total chlorophyll, and carotenoid contents in *P. frutescens*, whereas 1 µmol/L ABA had no significant effect on all of the photosynthetic pigment contents. These results are consistent with results from previous studies [34–36], indicating that ABA increases the photosynthetic pigment content in *P. frutescens*.

The antioxidant system in plants provides sufficient protection against the reactive oxygen species (ROS) damage under normal physiological conditions. Notably, increase in levels of ROS caused by adversity leads to imbalance of physiological metabolism in plants [37]. Exogenous ABA promotes activities of antioxidant enzymes (CAT, SOD, POD) in wheat to varying degrees, with the most significant effect observed on SOD [38]. In the present study, application of exogenous ABA increased the SOD and POD activities of *P. frutescens* to some extent, which can be attributed to upregulation of the expression of antioxidant enzyme genes such as Mn-SOD and peroxidase genes by ABA [39]. Notably, only 5 and 10 µmol/L ABA increased the CAT activity of *P. frutescens*, whereas 1 and 20 µmol/L ABA had no significant effect. This indicates that the effect of ABA on expression of different antioxidant enzyme genes may be different or multi-layered [40]. Soluble protein content plays an important osmoregulatory role in plant cells. It plays a key role in maintaining the structure and function of cells under stress conditions, as well as plays a stabilizing and protective role on the structure and function of biological macromolecules [38]. In the current study, 5, 10 and 20 µmol/L ABA increased the soluble protein content in *P. frutescens*, whereas 1 µmol/L ABA had no significant effect,

which is consistent with findings from a previous study [41]. This indicates that ABA induces formation of soluble proteins in *P. frutescens*. MDA is one of the products of membrane lipid peroxidation. It combines with proteins in the cell membrane resulting in cross-linking and polymerization of protein molecules [42] and accelerates degradation of chlorophyll [43]. MDA content is an important sign of stress damage to plants [44]. In the present study, ABA decreased the MDA content in *P. frutescens*, indicating that ABA effectively reduced membrane lipid peroxidation in *P. frutescens* and alleviated damage of the cell membrane. These findings indicate that ABA may improve resistance of *P. frutescens* to soil Se stress by increasing the antioxidant enzyme activity and reducing the membrane lipid peroxidation.

Absorption of Se by plants mainly depends on the environmental conditions, soil factors and the type of plants [45]. Selenite can be directly used by plants [46]. In the present study, 5 µmol/L ABA increased all of the Se content, BCF, and Se extraction of *P. frutescens*, whereas 1, 10, and 20 µmol/L ABA decreased or had no significant effect. These results are similar to the results on the effects of ABA on cadmium content and cadmium accumulation in *Bidens pilosa* reported previously [31]. In addition, regression analysis showed that the ABA concentration had a linear regression relationship with the root Se content, shoot Se content, root Se extraction, and shoot Se extraction of *P. frutescens*. Correlation analyses also showed the Se extraction had significant correlation with the POD activity, CAT activity, and Se content. This implies that only the appropriate concentration (5 µmol/L) of ABA could promote the Se uptake and accumulation in *P. frutescens*, and the antioxidant enzyme activity could play an important role in the Se uptake and accumulation. However, the Se absorption mechanism of *P. frutescens* should be explored further. In the current study, use of different concentrations of ABA had no significant effects on TF and TAF of *P. frutescens*, indicating that ABA could not affect the Se transportation from roots to shoots of *P. frutescens*. The main reason may be that most of the selenite is converted to organic Se compounds and remains in plant roots. Therefore, only low amounts of Se are transported to plant shoots [47, 48].

## Conclusions

Application of 5 and 10 µmol/L ABA promoted the growth of *P. frutescens* as indicated by increase in biomass and photosynthetic pigment content. In addition, these concentrations of ABA improved the resistance of *P. frutescens* against ROS by increasing the SOD, POD activity and soluble protein content, with 5 µmol/L ABA showing the most significant effect. Administration of 5 µmol/L ABA increased Se content, Se accumulation and BCF of *P. frutescens*. This implies that 5 µmol/L ABA can be used to promote the growth of *P. frutescens* as well as the Se uptake and accumulation in *P. frutescens*.

## Supporting information

**S1 Data.**
(XLS)

## Author Contributions

**Conceptualization:** Renyan Liao.

**Data curation:** Renyan Liao.

**Investigation:** Renyan Liao.

**Supervision:** Renyan Liao.

**Writing – original draft:** Renyan Liao.

**Writing – review & editing:** Renyan Liao.

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
