## [Decision Letter · Decision Letter 0]

23 Aug 2022

PONE-D-22-20189Effects of abscisic acid on growth and selenium uptake in medicinal plant Perilla frutescensPLOS ONE

Dear Dr. Liao,

Thank you for submitting your manuscript to PLOS ONE. After careful consideration, we feel that it has merit but does not fully meet PLOS ONE’s publication criteria as it currently stands. Therefore, we invite you to submit a revised version of the manuscript that addresses the points raised during the review process.

We look forward to receiving your revised manuscript.

Kind regards,

Mayank Gururani

Academic Editor

PLOS ONE

Journal Requirements:

Reviewers' comments:

Reviewer's Responses to Questions

**Comments to the Author**

1. Is the manuscript technically sound, and do the data support the conclusions?

Reviewer #1: Yes

Reviewer #2: Yes

2. Has the statistical analysis been performed appropriately and rigorously? 

Reviewer #1: Yes

Reviewer #2: Yes

3. Have the authors made all data underlying the findings in their manuscript fully available?

Reviewer #1: Yes

Reviewer #2: Yes

4. Is the manuscript presented in an intelligible fashion and written in standard English?

Reviewer #1: Yes

Reviewer #2: Yes

5. Review Comments to the Author

Reviewer #1: Well-designed and carried out research work has yielded beautiful and even useful and timely results for practice. Its main goals (see 2 p./26-27.) were achieved by the work. In addition, it has been demonstrated that „ABA may improve resistance of P. frutescens to soil Se stress by increasing antioxidant enzyme activity and reducing membrane lipid peroxidation.”

MS uses a relatively large number of abbreviations (ABA, SOD, POD, CAT, MDA, TF, TAF, BCF). Each of them receives an explanation at the first time of their application, but in different parts of the MS. Therefore, it may be suggested to create an Abbreviations chapter after the Keywords chapter, before the Introduction chapter. The inscriptions in Table 3 and Table 7 do not include explanations of abbreviations.

Ad 2 p. 27:

Please clarify the following thought/sentence: "to provide a basis for P. frutescens production. (production development ?, ?).

Letter and character errors:

ad 2nd p./ 24 (Corrected by Liao, 2018)

ad 2nd p./ 24 (Correct: Thlaspi arvense)

ad 12 p./ 5.

ad 13 p./ 16.

give. p. 13/ 13.

ad 14 p./ 7.

The Latin phrase Per capita (ad 1 p./29), as a text in a foreign language, is recommended to be highlighted, written in italics.

Literature:

It is recommended to distinguish between the two Li et al., 2019 citations (for example: a, with b notations)

They are included in the literature, but there is no citation in the text for the following two articles: Liu et al., 2009; Yi et al., 2020

Guignardi et Schiavon (2017): place of publication? (Cham?)

Latin name to highlight: p. 12 / 19.; 13. p./ 2 (chinensis)

Shao et al. citation does not seem complete.

Twice-featured citation: Wang et al., 2016.

Zhang et al. (2021) citation correct? In internet searches, the reviewer could not find it.

Reviewer #2: The tasks set correspond to the objectives of the study. The presented data allow us to assess the reliability of the work. The study was conducted qualitatively, but I think that such a result is expected. Concentrations of ABA can enhance the adaptation of plants to drought by inhibiting growth processes. Therefore, the authors did not receive a significant increase in growth characteristics. It would be interesting to consider the mechanism affecting the absorption of selenium. The research is interesting, but its significance is small. The work can be published because it may be of interest to young scientists.

6. PLOS authors have the option to publish the peer review history of their article (what does this mean?). If published, this will include your full peer review and any attached files.

Reviewer #1: No

Reviewer #2: **Yes: **Dr. Oksana Belous, Head of the Laboratory of Plant Physiology and Biochemistry of the Subtropical Scientific Center (Sochi, Russia)

---

## [Author Response · Author response to Decision Letter 0]

10 Sep 2022

Reviewers' comments:

5. Review Comments to the Author

Reviewer #1: Well-designed and carried out research work has yielded beautiful and even useful and timely results for practice. Its main goals (see 2 p./26-27.) were achieved by the work. In addition, it has been demonstrated that „ABA may improve resistance of P. frutescens to soil Se stress by increasing antioxidant enzyme activity and reducing membrane lipid peroxidation.”

Thank you for your reviewing.

MS uses a relatively large number of abbreviations (ABA, SOD, POD, CAT, MDA, TF, TAF, BCF). Each of them receives an explanation at the first time of their application, but in different parts of the MS. Therefore, it may be suggested to create an Abbreviations chapter after the Keywords chapter, before the Introduction chapter. The inscriptions in Table 3 and Table 7 do not include explanations of abbreviations.

We have added the Abbreviations section, and the abbreviations are also added in Table 3 and Table 7.

Ad 2 p. 27:

Please clarify the following thought/sentence: "to provide a basis for P. frutescens production. (production development ?, ?).

It should be production development. We have revised.

Letter and character errors:

ad 2nd p./ 24 (Corrected by Liao, 2018)

ad 2nd p./ 24 (Correct: Thlaspi arvense)

ad 12 p./ 5.

ad 13 p./ 16.

give. p. 13/ 13.

ad 14 p./ 7.

We have revised.

The Latin phrase Per capita (ad 1 p./29), as a text in a foreign language, is recommended to be highlighted, written in italics.

We have revised.

Literature:

It is recommended to distinguish between the two Li et al., 2019 citations (for example: a, with b notations)

We have revised.

They are included in the literature, but there is no citation in the text for the following two articles: Liu et al., 2009; Yi et al., 2020

We have deleted them.

Guignardi et Schiavon (2017): place of publication? (Cham?)

New York, USA. We have added.

Latin name to highlight: p. 12 / 19.; 13. p./ 2 (chinensis)

We have revised.

Shao et al. citation does not seem complete.

It may be the wrong citation. We have replaced by “Cui L, Zhao J, Chen J, Zhang W, Gao Y, Li Ba, Li YF (2018) Translocation and transformation of selenium in hyperaccumulator plant Cardamine enshiensis from Enshi, Hubei, China. Plant and Soil 425: 577-588. ”

Twice-featured citation: Wang et al., 2016.

We have deleted onr.

Zhang et al. (2021) citation correct? In internet searches, the reviewer could not find it.

It may be the wrong citation. We have replaced by “Sah SK, Reddy KR, Li J (2016) Abscisic acid and abiotic stress tolerance in crop plants. Frontiers in Plant Science 7:571.”

Reviewer #2: The tasks set correspond to the objectives of the study. The presented data allow us to assess the reliability of the work. The study was conducted qualitatively, but I think that such a result is expected. Concentrations of ABA can enhance the adaptation of plants to drought by inhibiting growth processes. Therefore, the authors did not receive a significant increase in growth characteristics. It would be interesting to consider the mechanism affecting the absorption of selenium. The research is interesting, but its significance is small. The work can be published because it may be of interest to young scientists.

Thank you for your reviewing.

---

## [Decision Letter · Decision Letter 1]

26 Sep 2022

Effects of abscisic acid on growth and selenium uptake in medicinal plant Perilla frutescens

PONE-D-22-20189R1

Dear Dr. Liao,

We’re pleased to inform you that your manuscript has been judged scientifically suitable for publication and will be formally accepted for publication once it meets all outstanding technical requirements.

Kind regards,

Mayank Gururani

Academic Editor

PLOS ONE

Additional Editor Comments (optional):

Reviewers' comments:

Reviewer's Responses to Questions

**Comments to the Author**

1. If the authors have adequately addressed your comments raised in a previous round of review and you feel that this manuscript is now acceptable for publication, you may indicate that here to bypass the “Comments to the Author” section, enter your conflict of interest statement in the “Confidential to Editor” section, and submit your "Accept" recommendation.

Reviewer #1: All comments have been addressed

2. Is the manuscript technically sound, and do the data support the conclusions?

Reviewer #1: Yes

3. Has the statistical analysis been performed appropriately and rigorously? 

Reviewer #1: Yes

4. Have the authors made all data underlying the findings in their manuscript fully available?

Reviewer #1: Yes

5. Is the manuscript presented in an intelligible fashion and written in standard English?

Reviewer #1: Yes

6. Review Comments to the Author

Reviewer #1: The author is to be thanked for accepting all the criticisms and suggestions of the reviewers.

The reviewer checked the changes indicated in the author's answer and found them to be in order.

These, including the new references, significantly increase the quality of the manuscript.

Some further minor formal errors:

- ad p. 11/19: correctly Arabidopsis.

-there is a space before the page numbers or there is no space (not uniform).

7. PLOS authors have the option to publish the peer review history of their article (what does this mean?). If published, this will include your full peer review and any attached files.

Reviewer #1: No

---

## [Editor Report · Acceptance letter]

28 Sep 2022

PONE-D-22-20189R1 

Effects of abscisic acid on growth and selenium uptake in medicinal plant *Perilla frutescens*

Dear Dr. Liao:

I'm pleased to inform you that your manuscript has been deemed suitable for publication in PLOS ONE. Congratulations! Your manuscript is now with our production department. 

Kind regards, 

on behalf of

Dr. Mayank Gururani 

Academic Editor

PLOS ONE